# Single-Cell RNA-Seq Analysis Reveals Lung Epithelial Cell Type-Specific Responses to HDM and Regulation by Tet1

**DOI:** 10.3390/genes13050880

**Published:** 2022-05-14

**Authors:** Tao Zhu, Anthony P. Brown, Lucy P. Cai, Gerald Quon, Hong Ji

**Affiliations:** 1California National Primate Research Center, University of California, Davis, CA 95616, USA; taozhu@ucdavis.edu (T.Z.); apbrown@ucdavis.edu (A.P.B.); lpcai@ucdavis.edu (L.P.C.); 2Department of Molecular and Cellular Biology, Genome Center, University of California, Davis, CA 95616, USA; gquon@ucdavis.edu; 3Department of Anatomy, Physiology and Cell biology, School of Veterinary Medicine, University of California, Davis, CA 95616, USA

**Keywords:** allergic lung inflammation, Tet1, single-cell RNA-seq, alveolar type 2 (AT2) cells, ciliated cells

## Abstract

Tet1 protects against house dust mite (HDM)-induced lung inflammation in mice and alters the lung methylome and transcriptome. In order to explore the role of Tet1 in individual lung epithelial cell types in HDM-induced inflammation, we established a model of HDM-induced lung inflammation in Tet1 knockout and littermate wild-type mice, then studied EpCAM^+^ lung epithelial cells using single-cell RNA-seq analysis. We identified eight EpCAM^+^ lung epithelial cell types, among which AT2 cells were the most abundant. HDM challenge altered the relative abundance of epithelial cell types and resulted in cell type-specific transcriptomic changes. Bulk and cell type-specific analysis also showed that loss of Tet1 led to the altered expression of genes linked to augmented HDM-induced lung inflammation, including alarms, detoxification enzymes, oxidative stress response genes, and tissue repair genes. The transcriptomic regulation was accompanied by alterations in TF activities. Trajectory analysis supports that HDM may enhance the differentiation of AP and BAS cells into AT2 cells, independent of Tet1. Collectively, our data showed that lung epithelial cells had common and unique transcriptomic signatures of allergic lung inflammation. Tet1 deletion altered transcriptomic networks in various lung epithelial cells, which may promote allergen-induced lung inflammation.

## 1. Introduction

Asthma is one of the most common pulmonary disorders with high heterogeneity [1]. Previous studies have noted transcriptomic differences in both asthmatic patients and animals with allergic lung inflammation [2,3,4]. Recent studies, including ours [2,5,6,7], have linked DNA methylation-associated transcriptomic changes to the pathogenesis and progression of asthma. Other studies have linked enzymes involved in DNA methylation maintenance to asthma pathogenesis [8]. Our studies in pediatric nasal epithelial samples and bronchial epithelial cells (HBECs) [9,10,11] suggest that Tet1 (Tet Methylcytosine Dioxygenase 1) may regulate childhood asthma and responses to environmental exposures. Tet1 is a ten-eleven translocation methylcytosine dioxygenase. These demethylases catalyze the hydroxylation of DNA methylcytosine (5mC) into 5-hydroxymethylcytosine (5hmC), 5-formylcytosine (5fC), and 5-carboxycytosine (5caC), interact with histone modifiers and transcription factors, and contribute to many biological processes and responses to environmental exposures (reviewed in [12]). In a follow-up study using genetic knockout mice [9], we found that Tet1 plays a protective role in allergic lung inflammation. Tet1 knockout mice exhibited significantly increased IL13 and IL33 (mRNA and protein levels) in the lungs. Concurrent with increased lung inflammation following house dust mite (HDM) challenges, Tet1 knockout mice showed a dysregulated expression of genes in several signaling pathways, including NRF2-mediated oxidative stress response, aryl hydrocarbon receptor (AhR) signaling, and interferon (IFN) signaling. These genes were also regulated by Tet1 in HBECs [9].

Given the use of RNA from total mouse lung tissues in these previous studies, critical questions remained regarding the cellular origin(s) of inflammatory mediators and molecules (e.g., IL33) following allergen challenges, and the role that Tet1 plays in regulating these mediators/molecules in individual cell types in vivo. In this study, our primary goal was to identify epithelial cell types that were influenced by HDM treatment and Tet1 knockout in vivo using single-cell RNA-seq analysis. We hypothesized that Tet1 regulates genes in proinflammatory and oxidative stress response pathways in specific lung epithelial cells, and that the loss of Tet1 has similar effects on these pathways compared to HDM treatment. The transcriptomic effects of Tet1 loss and HDM challenges were compared in all EpCAM^+^ cells together and in each cell type, and the effects of Tet1 loss in the presence of HDM was also evaluated. We observed similar effects of Tet1 deficiency and HDM on the transcription of alarmins and detoxification enzymes, as well as of the down-regulation of oxidative stress pathways and the upregulation of markers of epithelial injury following Tet1 knockout in the presence of HDM. These data suggest possible pathways through which Tet1 protects against HDM-induced lung inflammation. 

## 2. Methods

### 2.1. Establishment of HDM-Induced Airway Inflammation Mouse Model and Isolation of EpCAM^+^ CD31^−^ CD45^−^ (EpCAM^+^) Lung Epithelial Cells

HDM-induced allergic inflammation was established in both Tet1^−/−^ mice (knockout/KO mice) and their littermate Tet1^+/+^ mice (wild-type/WT mice), as described in a previous study [9] (details in Appendix A). All animal procedures were approved by the Institutional Animal Care and Use Committee at the University of California, Davis. The mice were divided into 4 groups, WT-Saline, WT-HDM, KO-Saline, and KO-HDM, with 2 mice in each. Airway inflammation was assessed by histology. Lobes, about 5 mm large, were sectioned in half and collected from each animal. Lobes from animals of the same group were pooled. Tissues were minced in a collagenase (480 U/mL)–dispase (5 U/mL)–DNase (80 U/mL) digestion solution in DPBS with Ca^+^ and Mg^2+^ and razor blades. The finely minced tissues were incubated in 6mL of digestion solution at 37 °C for 40 min, with gentle inversion for 30 s at 5 min intervals. After incubation, samples were washed with a DPBS/1% BSA wash buffer and passed through a 70 μm filter strainer. After spinning down cells and removing the supernatant, cells were treated with AKC lysis buffer to remove red blood cells, and washed once more to remove dead cells and excess RNA in the buffer. CD45^+^ cells (immune cells) and CD31^+^ (endothelial cells) were then depleted using mouse CD45 and CD31 MicroBeads (Miltenyi Biotec), and the remaining cells were counted and incubated with anti-CD326-APC (eBioscience), anti-CD31-PE-cy7 (eBioscience), and anti-CD45-PE (Biolegend). CD326^+^ CD31^−^ CD45^−^ cells were collected on a MoFlo sorter (Beckman Coulter/MoFlo XDP) with >99% purity at the Research Flow Cytometry Core at Cincinnati Children’s Hospital Medical Center (Appendix A). 

### 2.2. Single-Cell Library Preparation and Sequencing

For 10X Genomics single-cell RNA sequencing, cells were brought to a concentration of about 1000 cells/µL using the resuspension buffer. A total of 8000 cells were targeted for cDNA preparation and put through the 10X Genomics Chromium Next GEM Single Cell 3′ v3.1 (Pleasanton, CA, USA) pipeline for library preparation. After cDNA preparation and GEM generation and clean-up, cDNA was quality checked and quantified on an Agilent Bioanalyzer DNA High-Sensitivity chip at a 1:10 dilution. Successful traces showed a fragment size range of approximately 1000–2000 bp. A total of 12 indexing cycles for the library were used. The quality of the indexed libraries was checked on an Agilent Bioanalyzer DNA High-Sensitivity chip at a 1:10 dilution. Successful libraries had fragment size distributions with peaks around 400bp. Libraries were sequenced using 4 Illumina Hiseq lanes (Paired-end 75 bp).

### 2.3. Single-Cell RNA-Sequencing Data Analysis

We used Cell Ranger v3.1.0 [13] for initial processing of the scRNA-seq data. Specifically, we used Cell Ranger for sample demultiplexing, barcode processing, read alignment to the reference genome, data aggregation, and generating expression matrices. We then imported the data into Seurat v3.1.4 [14] for downstream processing. Cells that had fewer than 700 genes expressed, greater than 10% of reads mapping to the mitochondria, fewer than 20,000 unique molecular identifiers, or that expressed Pecam1 (CD31, an endothelial marker [15]) were filtered out. We also used cell cycle markers to identify which part of the cell cycle each cell was in during sampling [16], and subsequently regressed out this variable and the percentage of mitochondrial reads in each cell while normalizing and scaling the data with Seurat. In order to cluster all cell types accurately without biases from the HDM treatment, we used scAlign [17], a program that uses an unsupervised deep learning algorithm to correct for treatment effects that can affect cell clustering. First, the data were separated into two separate files based on treatment, and 3000 variable features were identified in each file separately. Variable features that overlapped were used for downstream clustering. We then performed single-cell alignment by performing a bidirectional mapping between cells with different treatments, creating a low-dimensional alignment space where cells were grouped together by type rather than by treatment [17]. Our best results (i.e., greatest separation of clusters by type rather than by treatment) were achieved by using the first 50 principal component scores, rather than individual gene-level data, as model inputs. The output from the scAlign model was subsequently used as the input for a uniform manifold approximation and projection (UMAP) in Seurat [14] to visualize the cell clusters. 

We used the FindClusters function from Seurat to cluster the cells, using a wide range of resolutions, from 0.01 to 2, to identify the best resolution for our dataset. FindClusters is a shared nearest-neighbor modularity optimization-based clustering algorithm. Our eventual selection was a resolution of 1, yielding 11 initial cell clusters. We then used markers from the single-cell mouse cell atlas [18] and another study on lung epithelial cells [19] to identify cell clusters. For each cluster, we plotted the gene expression levels of the markers from each epithelial cell type in our references, and identified which cell type was represented by each cluster. We merged some of the original 11 clusters identified by Seurat [14] that expressed similar markers, only merging sister nodes in the cell cluster tree generated by BuildClusterTree from Seurat. After merging similar clusters, we had a total of nine different labeled cell clusters in our dataset. We then used Seurat to calculate cluster means and perform differential expression analyses via the Wilcoxon rank sum test. For each individual cell type, as well as for all cells combined, we performed five differential expression comparisons: (1) WT-HDM vs. WT-Saline (WT-Sal), (2) KO-Saline (KO-Sal) vs. WT-Sal, (3) KO-HDM vs. KO-Sal, (4) KO-HDM vs. WT-HDM, and (5) KO-HDM vs. WT-Sal. Differentially expressed genes (DEGs) had a false discovery rate (FDR) of 0.05 or less, and a fold change of at least 1.2. DEGs also had to be expressed in at least ten percent of one of the comparison groups to qualify as differentially expressed. Pathway enrichment analyses for DEGs from each comparison were performed using IPA (QIAGEN Inc. Germantown, MD, USA, https://digitalinsights.qiagen.com/products-overview/discovery-insights-portfolio/analysis-and-visualization/qiagen-ipa/) (accessed between 1 March 2021 and 1 April 2022).

### 2.4. Transcription Factor Activity Analysis

We used DoRothEA [20,21,22] to assess the relative transcription factor (TF) activity levels in different cell types and/or treatments. DoRothEA computes activity levels for transcription factors using the expression levels of targets of each TF, instead of the expression level of the TF itself. For our analysis, we chose to include only highly confident interactions between TFs and target genes (confidence levels “A”, “B”, or “C”). We used VIPER for statistical analysis of the TF activity levels, as VIPER considers the mode of each TF–target interaction and has been shown to be appropriate for single-cell analysis [22,23]. We compared the TF activity between cell types and plotted the top 50 by activity level variance. We also compared the TF activity within cell types by sample type in AT2 and ciliated cells, and plotted the top 25.

### 2.5. Single-Cell Trajectory Analysis

In order to assess the differentiation of AP and BAS cells to AT2 cells, we performed single-cell trajectory analysis using Monocle [24]. First, the dataset was divided to include only AP, BAS, and AT2 cells using Seurat [14]. Next, cells were ordered based on the progress they had made through the differentiation process via reversed graph embedding using Monocle; then, dimensionality reduction was performed using the “DDRTree” reduction method, and cell trajectories were visualized. We also performed trajectory analyses on different subsets of AP, BAS, and AT2 cells to better understand how HDM treatment and Tet1 deletion impacted the differentiation of AP and BAS cells into AT2 cells. We performed cell trajectory analyses on (1) all AP, BAS, and AT2 cells; (2) only wild-type AP, BAS, and AT2 cells; (3) only AP, BAS, and AT2 cells that had Tet1 knocked out; (4) only saline-treated AP, BAS, and AT2 cells; (5) only HDM-treated AP, BAS, and AT2 cells. For each of these comparisons, we also performed branched expression analysis modeling within Monocle to identify genes that were differentially expressed along the trajectory at branch points that roughly separated AP, BAS, and AT2 cells. However, this was not possible in the saline-cell-only group, as there were no branch points that clearly separated these different groups of cells. We then used Monocle to make heatmaps, to visualize the expression patterns of genes that showed a significant change in expression along these trajectories (*q* < 1 × 10^−4^).

## 3. Results

### 3.1. Single-Cell Analysis Identified Nine Cell Types in the EpCAM^+^ CD45^−^ CD31^−^ Cells from Mouse Lungs

To identify cell types that are important in responses to HDM challenges and that are subject to Tet1 regulation, we isolated EpCAM^+^ CD45^−^ CD31^−^ (referred to as EpCAM^+^) lung epithelial cells from animals in an established HDM-induced allergic airway inflammation model [9] and characterized them using single-cell RNA-seq (overall study design in Figure 1A, treatment protocol and isolation strategy in Appendix A). To demonstrate the successful establishment of allergen-induced lung inflammation, pathological alterations were observed. Consistent with previous observations [9], airway inflammatory cell infiltration, goblet cell proliferation, smooth muscle hypertrophy, and airway basement membrane thickening were observed in HDM-challenged WT (WT-HDM) mice included in the single-cell analysis (Appendix A). Slightly more inflammatory cell infiltration, goblet cell proliferation, and airway mucus secretion were observed in HDM-challenged KO mice (KO-HDM) that were included in this scRNA-seq analysis. No pathological changes were observed in saline-treated mice.

Among the 25,437 cells that passed our quality filters, we identified 9 cell types (Figure 1B,C) based on established marker expression (Figure 1D, Appendix A). As expected, most of the cell types expressed the respiratory epithelial cell markers *Epcam* and *Scgb1a1* (Appendix A). Interestingly, Tet1 was found to be mostly expressed in AT2 cells, alveolar progenitor cells, bronchial stem cells, ciliated cells, and club cells. In order of abundance, we identified AT2, alveolar progenitor (AP), broncho alveolar stem (BAS), ciliated, stromal, club, goblet, neuroendocrine, and AT1 cells (Figure 2 and Appendix A). On average, we observed a 24.9% increase in AP cells and an 11.1% increase in BAS cells following HDM challenges, while there was a 29.2% and 3.1% decrease in AT2 and ciliated cells, respectively, following HDM challenges (Figure 2 and Appendix A). There was also a small increase in the percentage of goblet cells following HDM challenges, consistent with goblet cell metaplasia in histological studies (Appendix A). Although we expected cell loss during single-cell preparation, especially for fragile AT1 cells, these results suggest dynamic changes in lung epithelial cell proportions in HDM-induced lung inflammation. Additionally, since stromal cells are not epithelial cells, we will not discuss DEGs in stromal cells further.

### 3.2. Exposure to HDM Led to Cell Type-Specific Changes in the EpCAM^+^ Lung Epithelial Cells

We identified 830 differentially expressed genes (DEGs) between WT-HDM and WT-Sal cells in our analysis of all EpCAM^+^ cells (hereafter referred to as bulk cell analysis), regardless of cell type (Table 1 and Appendix A). Based on our pathway analysis (IPA) results, several pathways were significantly activated or inhibited, including oxidative phosphorylation (activation z-score = 7.353), sirtuin signaling pathway (z-score = −2.897), and xenobiotic metabolism AHR signaling pathway (z-score = −2.111) (Appendix A). We further identified DEGs within each of the eight cell types in the WT-HDM vs. WT-Sal comparison. In total, we found 1773 DEGs within individual cell types (Table 1). All cell types except AT1 had differential expression in the WT-HDM vs. WT-Sal comparison. Of these 1773 DEGs, 616 were unique to one cell type (Figure 3). AT2 cells had the most DEGs associated with HDM treatment (*n* = 732), followed by AP cells (*n* = 431) (Table 1 and Appendix A). Many genes, including *Il33* [25], *Cxcl5* [26], *Cxcl15* [27], *Ccl20* [28], *Cxcl3* [29], *Cxcl1* [30], *C3* [31], and *Pfn1* [32], which have been linked to asthma pathogenesis, showed a cell type-specific response to HDM (Figure 4A,B, Appendix A). Subsequently, pathway analysis identified 120 significantly enriched pathways (Appendix A). While 98 of these pathways were shared with the bulk cell analysis, 22 pathways were unique to AT2 cells, including ferroptosis signaling pathway, chemokine signaling, and LPS-stimulated MAPK signaling.

Ciliated cells are one of the major target cells in asthma [33,34], and injured ciliated cells release inflammatory mediators to aggravate allergen-induced airway inflammation [35,36]. There were 210 DEGs in ciliated cells in the WT-HDM vs. WT-Sal comparison (Figure 3A and Figure 4C,D and Appendix A). Some DEGs from this comparison, such as *Fos* [37], *Jun* [38], *Hes1* [39], *Cd14* [40], and *C3* [31], have been associated with asthma pathogenesis. Subsequently, we identified 96 significantly enriched pathways in these DEGs (Appendix A). NRF2-mediated oxidative stress response was predicted to be inhibited (z-score = −2.333). Furthermore, 18 IPA pathways were enriched in the 104 unique DEGs in ciliated cells, including NRF2-mediated oxidative stress response (Appendix A). Collectively, these results suggest that different types of EpCAM^+^ lung epithelial cells responded to HDM and may contribute to airway inflammation, with common features as well as unique functions.

### 3.3. Tet1 Knockout Led to Cell-Specific Changes in Baseline EpCAM^+^ Lung Epithelial Cells

In the bulk cell analysis comparing KO-Sal and WT-Sal cells, 98 genes were differentially expressed (Table 1 and Appendix A). There were 27 enriched pathways associated with these DEGs (Appendix A), including several stress-related pathways (protein ubiquitination pathway/unfolded protein response/eNOS signaling/NRF2-mediated oxidative stress response/glutathione redox reactions I), and pro-inflammatory pathways (interferon signaling/IL-6 signaling/IL-12 signaling and production in macrophages). 

Furthermore, we identified Tet1 loss-associated DEGs in each cell type (Table 1 and Appendix A). AT2 and ciliated cells had the most DEGs. There were 70 such DEGs in AT2 cells, including 48 DEGs unique to AT2 cells (Appendix A). There were 45 enriched IPA pathways associated with these DEGs (Appendix A), including 28 unique to AT2 cells (e.g., AhR signaling/xenobiotic metabolism AHR signaling pathway). Several of these DEGs associated with Tet1 loss in AT2 cells (*Il33* [25], *Areg* [41] and *Bpifa1* [42], *Hspa8* [43] and *Lgals3* [44] (also DEGs in ciliated cells)), have been previously linked to asthma. Additionally, 41 Tet1 loss-associated DEGs were found in ciliated cells (Table 1 and Appendix A). Several asthma-related genes were identified: detoxification enzymes (*Cyp2a5* [45], *Gsto1* [46], *Gstp1* [47], *Gstm1* [48], and *Aldh1a1* [49]) were downregulated while *Hmgb1* (alarmin) [50,51] was upregulated (Figure 4C, Appendix A). Consistent with our observations following HDM treatment, the downregulation of the detoxification enzymes following Tet1 KO (*Gstp1*, *Gstm1*, and *Gpx2*) were only found in ciliated cells (Figure 4C,D and Appendix A). *Hmgb1* was also upregulated in AT2 cells following Tet1 KO. We identified 49 enriched pathways in these DEGs in ciliated cells (Appendix A), including a significant inactivation of the xenobiotic metabolism AHR signaling pathway (z-score = −2.449) and an enrichment of several other stress response pathways.

In summary, we found that Tet1 loss in saline-treated mice led to (1) increased expression of alarmins that promote type 2 inflammation, particularly *Il33* (AT2 cells) and *Hmgb1* (AT2 and ciliated cells), (2) the upregulation of genes that promote the proliferation and migration of airway smooth muscle cells and tissue remodeling (*Malat1* (AT2 cells) and *Pfn1* (six cell types)), and (3) the reduced expression of detoxification enzymes (*Gpx2*, *Gstm1*, *Gsto1*, and *Gstp1*) involved in AhR signaling in ciliated cells. Together, this gene dysregulation following Tet1 loss may have resulted in a pro-inflammatory transcriptomic state in AT2 and ciliated cells.

### 3.4. Tet1 Knockout and HDM Exposure Induced Overlapping Cell Type-Specific Changes in EpCAM^+^ Lung Epithelial Cells

To further test the hypothesis that the loss of Tet1 generates a proinflammatory transcriptomic state similar to that of HDM, we searched for overlapping genes among the WT-HDM vs. WT-Sal, KO-Sal vs. WT-Sal, and KO-HDM vs. WT-Sal comparisons. In the bulk cell analysis, we identified 48 overlapping DEGs, including 36 DEGs where Tet1 loss and HDM treatment led to changes in the same direction (Table 1 and Appendix A). Thirteen of these overlapping DEGs have been linked to asthma, including eight upregulated genes (*Il33, Hmgb1*, *Ager*, *Chia1*, *Ifitm3, Igfbp7*, *Pfn1*, *Retnla*) and five downregulated genes (*Atf3*, *Btg2*, *Klf6*, *Neat1*, and *Sec14l3*) (Figure 4A,B and Appendix A). Eight mitochondrially encoded genes were also among these thirty-six overlapping DEGs, but their role in asthma is not clear. Next, overlapping DEGs within each cell type were identified and investigated further. In AT2 cells, there were 39 shared DEGs in all 3 different comparisons in AT2 cells (Appendix A). Of these 39 DEGs, 30 were affected similarly by Tet1 loss and/or HDM treatment. Meanwhile, there were five different genes that were consistently downregulated in ciliated cells following Tet1 loss and/or HDM treatment (Appendix A), including the asthma-associated genes *Gstp1*, *Gstm1*, and *Gpx2* (Figure 4C,D). Additionally, we found that two genes (*Dmkn* and *Chia1*) in AP cells and one gene (*Nnat*) in BAS cells were consistently changed in a similar fashion. Combining all these genes with the same directions of changes in EpCAM+ cells, AT2 cells, AP, BAS, and ciliated cells, there was significant enrichment for NRF2-mediated Oxidative Stress Response, HMGB1 Signaling and Airway Inflammation in Asthma (Appendix A). IPA analysis also revealed several significant diseases and functions affected by these overlapping genes, including organ inflammation in EpCAM^+^ cells and AT2 cells (Figure 5A,B), cell proliferation of fibroblast in AT2 cells (Figure 5C), and the synthesis of reactive oxygen species in ciliated cells (Figure 5D). Collectively, these data support our hypothesis and suggest that the effects of Tet1 knockout in promoting allergen-induced lung inflammation were cell-type dependent.

### 3.5. Tet1 Dysregulates the Single-Cell Transcriptomic Signature of HDM-Induced Lung Inflammation in Mice

To further understand the effects of Tet1 loss in the presence of HDM, we studied the 72 genes that were differentially expressed in KO-HDM compared with WT-HDM in all EpCAM^+^ cells (Appendix A). Among the 72 DEGs, 29 significantly enriched pathways were identified (Appendix A), including several stress response pathways and proinflammatory pathways. This is consistent with our previous observations in bulk RNA-seq from the total lungs in mice challenged using the same exposure protocol [9]. Genes involved in tissue injury/repair and remodeling, such as *Areg* [52,53], *Malat1* [54,55], *S100a1* [56,57], *Mt1/2* [58,59], *Scgb1a1* [60,61], and *Clu* [62,63] (Appendix A), were upregulated in KO-HDM compared to WT-HDM, even though some of these genes (e.g., *Areg*) were downregulated in KO-Sal mice compared to WT-Sal mice.

We then compared KO-HDM with WT-HDM in each cell type. Interestingly, of the 195 DEGs identified across all cell types, 194 were identified in AP cells, AT2 cells, or BAS cells (Table 1 and Figure 3D), and 32 DEGs were shared among all 3 of these cell types. All 32 of these common DEGs changed in the same direction: 16 were upregulated (e.g., *Areg*, *Malat1*, *Mt1*/*2*, and *Scgb1a1*) and 16 were downregulated (e.g., *Lrg1*, *Pfn1*, and *Rnase4*) in these 3 cell types following Tet1 deletion (Appendix A). In AT2 cells, specifically, there were 78 DEGs in KO-HDM compared to WT-HDM, including several genes previously linked to asthma [43,64,65,66] (Appendix A). Further, we identified 32 significantly enriched pathways in the DEGs from AT2 cells (Appendix A), including stress response pathways (unfolded protein response/endoplasmic reticulum stress pathway/protein ubiquitination pathway/NRF2-mediated oxidative stress response (gene network shown in Figure 5E), in which most genes showed downregulation in KO-HDM. Not surprisingly, the same pathways were also significantly enriched among combined DEGs from all EpCAM^+^ cells, AT2 cells, AP, and BAS with the same directions of changes in the KO-HDM vs. WT-HDM comparison (Appendix A). Genes involved in apoptosis and cell viability were also found (Figure 5F,G). Similar to the bulk cell analysis, several chemokines and cytokines (e.g., *Cxcl5*) showed reduced expression in KO-HDM (Appendix A). In summary, our data suggest that, following HDM challenges, the loss of Tet1 led to the downregulation of genes in stress response pathways, potentially leading to cell apoptosis, tissue injury, and additional lung inflammation and remodeling.

### 3.6. Tet1 Knockout and HDM Exposure Induced Cell Type-Specific Changes in Transcription Factor (TF) Activity in EpCAM^+^ Lung Epithelial Cells

As Tet1 may regulate gene expression through interacting with TFs, TF activity in individual cell types was explored using DoRothEA [20,21,22]. Consistent with previous findings [67,68,69], we observed cell type-specific TF activity, consistent with their cell type-specific regulation of gene expression programs (Figure 6A). For example, *E2f1-4* and *Tfdp1* (both promote the expression of a cell cycle-related network [67]) were highly activated in BAS compared to all other cell types, while the activities of *Rest* (a repressor of neuronal genes in non-neuronal cells [70]) and *Rreb1* were lowest in neuroendocrine cells. Additionally, there was relatively high activity of *Rfx1* (promotes ciliogenesis [71,72]) in ciliated cells. Because AT2 cells were the most abundant cell type, there were generally only small changes in TFs activity compared to the average across all cells in our heatmap (Figure 6A). However, a group of TFs, including *Foxa1*, *Sox10*, and *Runx2*, showed relatively high activity levels in AT2 cells, and these TFs are essential in conducting airway and alveolar epithelium development and repair [73,74,75]. 

We also found that TF activity levels were altered by HDM and Tet1 loss in individual cell types, such as AT2 cells (Figure 6B–E) and ciliated cells (Figure 6F–H). For example, regardless of Tet1 status, *Foxa1* (also decreased in expression in HDM samples in AT2 cells, Appendix A) and *Hnf1a* activities were reduced by HDM in AT2 cells (Figure 6B), while the activities of *Rest, Rreb1*, and *Nr2c2* were increased by HDM in both AT2 and ciliated cells (Figure 6B,F). Meanwhile, *Smad3*, *Smad4*, and *Sp3* activities were decreased by HDM in ciliated cells (Figure 6F). When looking specifically for effects of Tet1, we found that the activities of *Myc, Rest*, and *Rreb1* were consistently reduced when Tet1 was lost in AT2 cells, regardless of HDM status (Figure 6C). Hif1α, whose expression is regulated by Tet1 [76] and may also regulate Tet1 expression during oxidative stress [77], showed reduced activity in KO-Saline (Figure 6D). Increased *Foxa1* activity was only observed in KO-HDM compared to WT-HDM (Figure 6E). In ciliated cells, however, nearly all effects of Tet1 loss were specific to the treatment state (Figure 6F). With saline treatment, activities of *Nfe2l2* (encoding Nrf2), *Nfe2l1* (encoding Nrf1) and their binding partner Mafk (Figure 6G) [78] were reduced in Tet1 KO cells compared to wild-type cells; these TFs are involved in various stress responses, including oxidative stress [79,80], and regulate the expression of ROS-detoxifying enzymes including *Gstp1*, *Gstm1*, and *Gpx2* [81]. Meanwhile, TFs involved in the aryl hydrocarbon receptor signaling pathway (AhR) and the TGFβ signaling pathway (Smad3, Smad4) had higher activity in KO-HDM compared to WT-HDM, even though HDM challenges reduced their activity (Figure 6F,H). Collectively, our data suggest that TFs respond to HDM challenges and Tet1 loss in a cell type-specific manner, which may contribute to cell type-specific transcriptomic changes that could be associated with lung inflammation.

### 3.7. HDM, Not Tet1, Promotes the Differentiation of AP and BAS Cells into AT2 Cells

As observed above, HDM challenges seemingly reduced AT2 cell proportions and increased AP and BAS cell proportions (Figure 2 and Appendix A). It has been established that AT2 cells can differentiate from AP and BAS cells [82,83]. Therefore, we performed single-cell trajectory analysis to explore the roles of HDM and/or Tet1 KO in AT2 cell differentiation (Appendix A). Initially, AT2, AP, and BAS cells from all groups were combined (Appendix A), and then cell fates were separated at branch point 4 (Appendix A). Our data support that AT2 cells differentiated from AP and BAS cells. Comparing WT-HDM to WT-Sal and KO-HDM to KO-Sal, HDM-challenged cells were clearly separated along the trajectory from saline-treated cells, indicating that HDM promotes AP and BAS proliferation (Appendix A). However, this was not observed in the KO-Sal vs. WT-Sal comparison, as no clear branch point was identified to separate cell fates, indicating that Tet1 KO did not have a substantial impact on AP and BAS proliferation (Appendix A). Thus, heatmaps showing significant changes along the trajectory were only generated for the other comparisons (Appendix A). *Sftpc* (a marker for AT2 and BAS cells), *Sftpa1* (a marker for AT2 and AP cells), and *Sftpb* (a marker for only AT2 cells) varied significantly along all trajectories, other than in the saline-only comparison (there were no comparable genes for this comparison due to a lack of clear branch points separating cell fates; Appendix A, Appendix A). Combined with evidence in Appendix A and Appendix A, these results suggest that HDM-induced AT2 cell injury promoted the proliferation and differentiation of AP and BAS cells. Tet1 knockout did not appear to affect the differentiation of AP and BAS cells into AT2 cells.

## 4. Discussion

The lung is an organ with continuous exposure to environmental challenges, such as allergens and pathogens. The epithelium is the first line of defense for the lung. Lung epithelium (airway epithelium [84] and alveolar epithelium [85,86]) injury and inflammation play critical roles in asthma. Our previous study [9] found that Tet1 deletion enhanced HDM-induced lung inflammation in mice, and that it was linked to transcriptomic changes in proinflammatory molecules (*Il33*) and detoxification enzymes (*Gsto1* and *Aldh1a1*) in total lung RNA. Furthermore, our previous results indicated that 59 pathways, including NRF2-mediated oxidative stress response, AhR signaling, and the interferon (IFN) signaling pathway, were significantly altered. However, the effects of Tet1 loss in specific lung cell types and the contribution of individual cell types in HDM-induced transcriptomic alterations remained unknown. Data from our current analyses showed that while AT2, alveolar progenitor cells, broncho alveolar stem cells, and ciliated cells underwent transcriptional changes following HDM challenges, AT2 and ciliated cells were the cell types most affected by Tet1 loss. Following HDM treatment, the upregulation of the alarms *IL33* and *Hmgb1* mainly occurred in AT2 cells, while the downregulation of detoxification enzymes mainly occurred in ciliated cells. Notably, Tet1 knockout had similar effects on alarms in AT2 cells and detoxification enzymes in ciliated cells. Further, in the presence of HDM, we found that genes involved in stress response pathways (e.g., *Xbp1*, *Dnajb9*, and *Hspa8*) and markers for tissue damage response/remodeling (e.g., *Clu*, *Mt1/2*) were upregulated. Based on these observations, a model linking Tet1 loss to increased airway inflammation was proposed (Figure 7). Although this model warrants further validations to establish the direct contributions of these pathways to Tet1-mediated inflammation, our data provide supporting evidence at the transcriptomic level. Future studies focusing on mechanisms through which Tet1 regulates these genes in the lung epithelium would also provide clarity on the regulatory process.

Some genes that were differentially expressed in our previous bulk RNA-seq analyses were not found in the current scRNA-seq analyses, and vice versa. It is possible that HDM-induced transcriptomic changes masked the effects of Tet1 knockout on lung epithelial cells in our current study. Meanwhile, increased noise in scRNA-seq data, lower power due to small cell numbers, the differences in how DEGs were defined in the scRNA-seq dataset compared to our previous RNA-seq studies, and the possible contribution of other lung non-epithelial cell types in bulk RNA-seq were also potential reasons for these inconsistencies. On a related note, in our current study, we observed the downregulation of several asthma-associated genes (e.g., *Cxcl5, Atf3* [87], and *Btg2* [88]) that are normally increased by HDM challenges, possibly due to the time-dependent expression pattern of these molecules during the acute phase of inflammation; this needs further investigation. In general, however, we noted a differential expression of several candidate genes in our scRNA-seq dataset that was consistent with prior studies. These include genes involved in oxidative stress responses and AhR signaling/detoxification, pro-inflammatory responses and alarmins, and tissue damage and repair, many of which are known to contribute to airway inflammation in asthma. 

AT2 cells were the most abundant cell type in our dataset (69.7% in total). Although no significant pathological alterations in alveoli were observed in either asthmatic patients or animal models, several studies [85,86,89] have reported that AT2 cells are a major contributor of inflammatory mediators in asthma. In particular, Ravanetti L. et al. supported that AT2 cells are one major source of IL-33 in mice with HDM-induced lung inflammation [89]. Our present study supports that HDM may enhance AT2 cell differentiation from AP and BAS cells (Appendix A), likely due to HDM-induced AT2 cell injury and death (Figure 2 and Appendix A). Our cell type-specific transcriptomic analysis showed that AT2 cells were one major source of inflammatory mediators and molecules, especially *Il33* (Figure 4A)*, Hmgb1*, *Ager* (Appendix A), and *Retnla* (Appendix A), in allergen-induced lung inflammation (Appendix A). Allergens may induce lung Th2 inflammation through AT2 cell injury, which releases alarms such as *Il33* [25] and *Hmgb1* [90], and Tet1 plays a critical role in this process by regulating their expression. Additionally, when comparing KO-HDM to WT-HDM in AT2 cells, we found that tissue damage and repair genes were upregulated, while oxidative stress response genes were downregulated; most of these genes were also DEGs in all EpCAM^+^ cells (Appendix A). The downregulation of oxidative stress response pathways, especially unfolded protein response (UPR), may result in apoptosis and cell death [91], and the roles of UPR in oxidative stress response, cytokine production, and asthma pathogenesis are emerging [92]. Taken together, our data suggest that while AT2 cells from Tet1 knockout mice at baseline have a pro-inflammatory transcriptomic signature (elevated alarms and reduced detoxification enzymes), HDM challenges may further induce cell damage and death due to elevated oxidative stress and downregulated stress responses. 

Ciliated cells are another major cell type within airway epithelium. Ciliated cells are essential for mucociliary clearance through their motile cilia’s coordinated directional movement, and can sense and respond to allergens, air pollution particles, and pathogens [33,93]. Asthma can cause cilia dysfunction and ultrastructural abnormalities, such as the decrease of ciliary beat frequency, the generation of dyskinetic and immotile cilia, and the disruption of tight junctions, which were closely associated with asthma severity [94]. Meanwhile, multiple studies [36,95] suggest that ciliated cells are a source of cytokines, such as IL33 and TSLP, in asthma. In our single-cell analysis, we only observed a trend of increase for *Il33* at baseline when Tet1 was deficient, and no further changes following HDM treatment. However, in vitro studies using HBECs showed that the loss of Tet1 significantly increased *IL33* expression following 24 h of HDM challenges (Appendix A), which support that Tet1 deficiency promotes the predisposition of allergic inflammation in ciliated cells. Further, we found that *Gstp1*, *Gstm1*, and *Gpx2* were all downregulated with HDM and/or Tet1 loss in ciliated cells (Figure 4C,D). These three genes (along with two others: *Ftl1* and *Nr4a1*) were consistently downregulated in ciliated cells, but not in AT2 cells (Appendix A), implying that there were unique responses to Tet1 loss and HDM treatment in ciliated cells. Our previous studies in HBECs showed that Tet1 regulates detoxification enzymes in HEBCs as well [9]. Together, these data highlight the unique role of ciliated cells and suggest that Tet1 may regulate HDM-induced lung inflammation via the regulation of AhR signaling/detoxification and pro-inflammatory cytokines/alarmins in ciliated cells (Figure 7).

Our data also suggest that HDM and/or Tet1 KO in mice lungs led to cell type-specific changes in TF activity levels (Figure 6). Our prior study using HBECs suggested that Tet1 protects against HDM-induced allergic inflammation at least partially by regulating the AhR pathway [9]. Data from the current study in ciliated cells support that AhR activity and the AhR pathway were altered following Tet1 KO and/or HDM treatment (Figure 6F,H and Figure 7). Meanwhile, Nrf1/2 activity was lower in KO-Saline mice, compared to WT-Saline (Figure 6G), which may explain the lower expression of *Gstp1*, *Gstm1*, and *Gpx2* in KO-Sal, compared to WT-Sal, as the Nrf2-mediated pathway is an alternative to the AhR pathway that regulates the expression of antioxidant enzymes [81]. In addition, consistent with an increased expression of markers for tissue damage and repair, the activity of Foxa1, a TF essential in airway epithelium development [96] and barrier integrity [75], was increased in KO-HDM compared to WT-HDM. Of note, these activity levels for the transcription factors were estimated using gene expression profiles of their targets, not the direct expressions of the transcription factor themselves. Even if the direct expression of a transcription factor were not altered, it is possible that Tet1 KO or HDM treatment could alter transcription factor activities, especially considering that Tet1 can directly interact with transcription factors [97,98]. Taken together, these results indicate that Tet1 plays an important role in regulating TF activity that may directly impact allergen-induced lung inflammation.

## 5. Conclusions

Collectively, our results revealed possible mechanisms underlying the protective role of Tet1 in HDM-induced lung inflammation at the single-cell level. Different epithelial cells played common and unique roles in allergic lung responses. The major cellular contributors of multiple asthma-associated inflammatory mediators, such as *Il33*, *Hmgb1*, *Retnla*, and *Ager* were identified, and a cluster of novel allergy-associated molecules, such as *Chia1*, *Nnat*, and *Nr4a1* were found in different lung epithelial cells. Future studies should attempt to elucidate the role of these novel targets in asthma. Our results suggest that AT2 cells are essential for Th2 inflammation through stress responses, alarms, tissue injury and repair in allergen-induced lung inflammation, and that these genes are regulated by Tet1. In ciliated cells, Tet1 promotes the Xenobiotic Metabolism AhR signaling pathway, especially the detoxification enzymes, which may protect the lung from allergen-induced inflammation. As chronic airway inflammation induced by allergens plays a critical role in the pathophysiology of asthma, these novel findings support that Tet1 is a potential therapeutic target of asthma.

## Figures and Tables

**Figure 1 genes-13-00880-f001:**
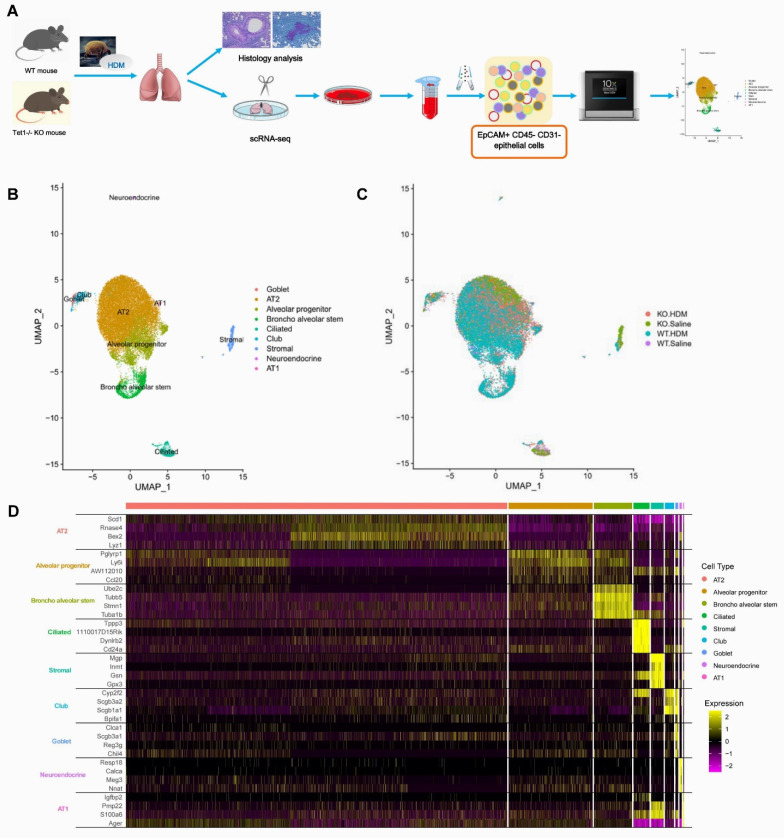
Single-cell RNA sequencing clustering analysis identifies 9 cell types in EpCAM^+^ CD45^−^ CD31^−^ lung cells. (**A**) Schematic overview of the study workflow. (**B**) UMAP visualization of clustering, revealing 9 cell populations. Population identities were determined based on marker gene expression. (**C**) UMAP visualization of EpCAM^+^ CD45^−^ CD31^−^ lung cells in Tet1 KO mice and WT mice with and without HDM challenge. (**D**) Heatmap of top 4 markers in 9 cell clusters.

**Figure 2 genes-13-00880-f002:**
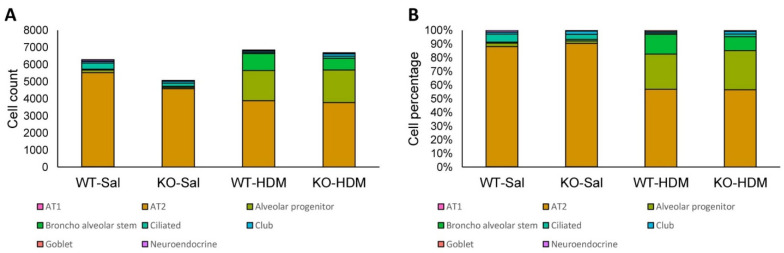
The abundance and percentage of each cell type. (**A**) Abundance of each cell type at each condition; (**B**) percentage of each cell type at each condition.

**Figure 3 genes-13-00880-f003:**
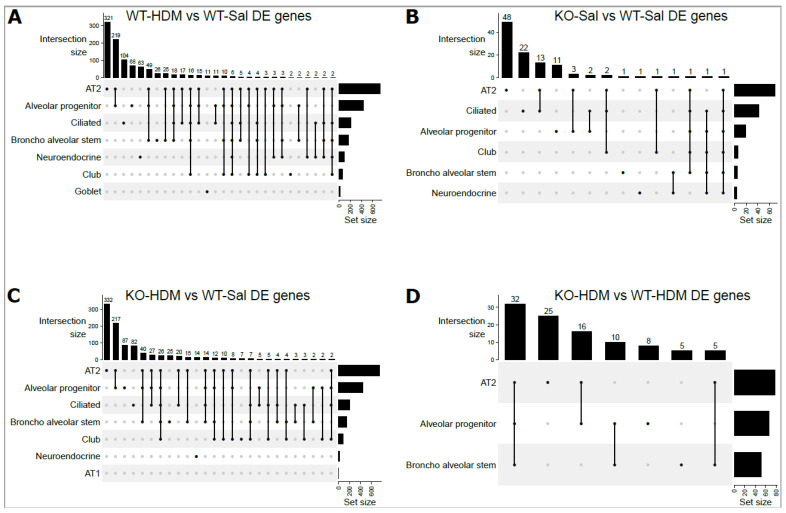
UpSet plots for overlapped DEGs among 8 individual cells types: (**A**) WT-Sal vs. WT-HDM; (**B**) WT-Sal vs. KO-Sal; (**C**) KO-HDM vs. WT-Sal; (**D**) KO-HDM vs. WT-HDM. Dots below vertical bars represent overlapping DEGs among different cell types. Horizontal bars represent the total number of DEGs in each cell type.

**Figure 4 genes-13-00880-f004:**
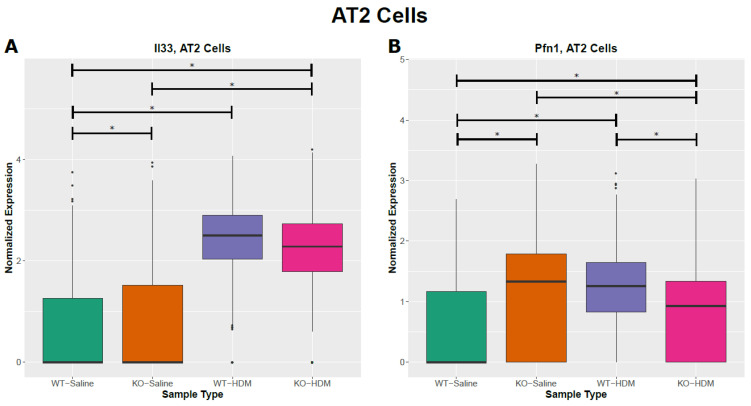
Cell-type specific changes resulting from HDM challenges and/or Tet1 deficiency. (**A**,**B**) AT2 cells; (**C**,**D**) ciliated cells. * represents FDR ≤ 0.05.

**Figure 5 genes-13-00880-f005:**
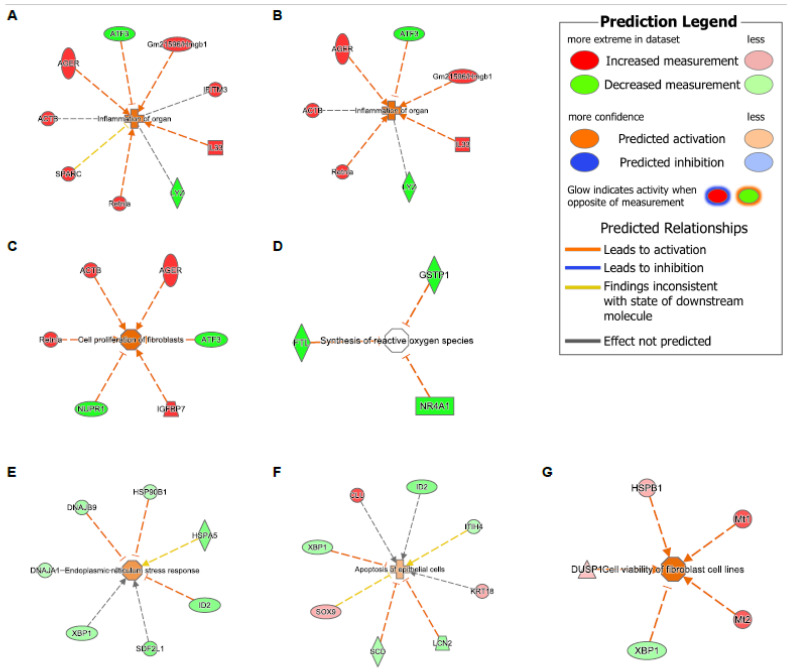
Diseases and function analysis in EpCAM^+^, AT2, and ciliated cells via IPA. (A-D) DEGs with consistent direction changes among 3 comparisons (WT-HDM vs. WT-Sal, KO-Sal vs. WT-Sal, and KO-HDM vs. WT-Sal). (**A**) Inflammation of organ in EpCAM^+^ cells (z = 1.866, overlapping *p* = 7.54 × 10^−4^). (**B**) Inflammation of organ in AT2 cells (z = 2.097, *p* = 2.93 × 10^−3^). (**C**) Cell viability of fibroblast in AT2 cells (z = 2.408, *p* = 3.51 × 10^−6^). (**D**) Synthesis of reactive oxygen species in ciliated cells (*p* = 3.54 × 10^−4^). (**E**) Endoplasmic reticulum stress response (z = 1.015, *p* = 6.74 × 10^−6^), (**F**) apoptosis of epithelial cells (z = 0.651, *p* = 6.85 × 10^−5^), and (**G**) cell viability of fibroblast cell lines (z = 2.236, *p* = 1.41 × 10^−4^) in DEGs in KO-HDM vs. WT-HDM in AT2 cells.

**Figure 6 genes-13-00880-f006:**
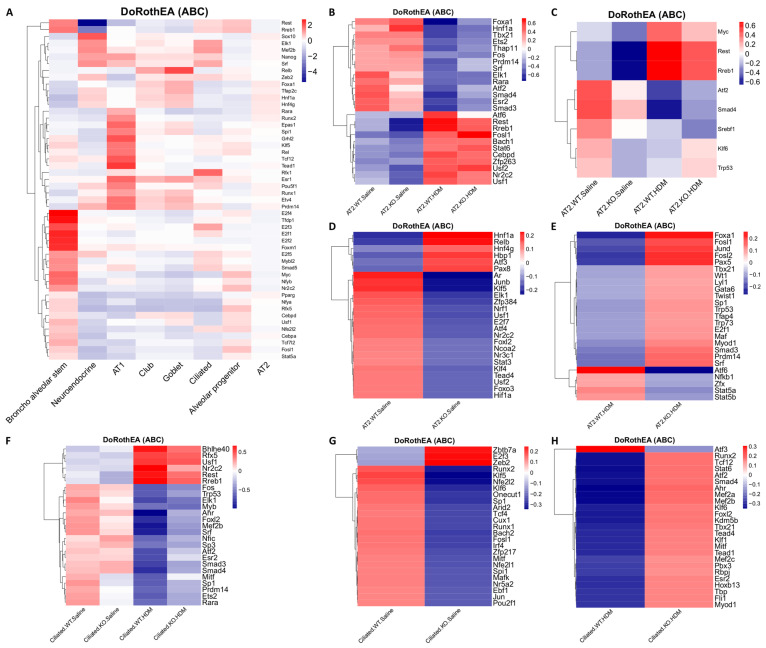
Heatmaps of transcription factor (TFs) activities analyzed using DoRothEA. (**A**) Top 50 most variable TFs by gene activity across 8 individual cell types. (**B**) Top 25 TFs across all AT2 cells. (**C**) TFs that were in the top 25 in both the KO-HDM vs. WT-HDM and KO-Saline vs. WT-Saline comparisons in AT2 cells. (**D**) Top 25 TFs in the KO-Saline vs. WT-Saline comparison that were not in the top 25 for KO-HDM vs. WT-HDM in AT2 cells. (**E**) Top 25 TFs in the KO-HDM vs. WT-HDM comparison that were not in the top 25 for KO-Saline vs. WT-Saline in AT2 cells. (**F**) Top 25 most-variable TFs in ciliated cells. (**G**) Top 25 most-variable TFs in ciliated cells in KO-Saline vs. WT-Saline. (**H**) Top 25 most-variable TFs in ciliated cells in KO-HDM vs. WT-HDM.

**Figure 7 genes-13-00880-f007:**
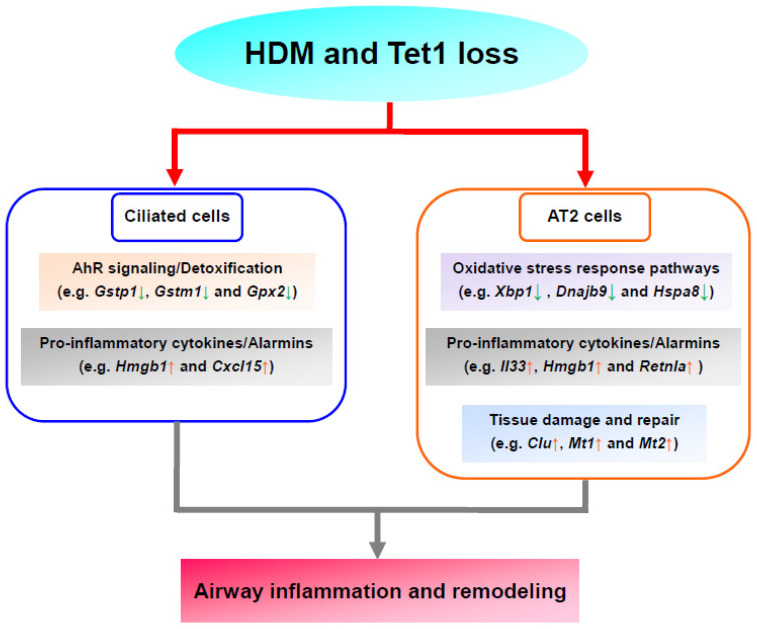
Proposed model based on transcriptomic analysis. Tet1 deletion downregulated (downward arrows in green) genes in AhR signaling/detoxification (ciliated cells, base line) and oxidative stress responses (AT2 cells, with HDM), and upregulated (upward arrows in red) genes in pro-inflammatory cytokines/alarmins (AT2 and ciliated cells, baseline) and tissue damage and repair (AT2 cells, with HDM). Collectively, this may aggravate HDM-induced airway inflammation and remodeling.

**Table 1 genes-13-00880-t001:** Number of differential expression genes (DEGs) of EpCAM^+^ cells.

**Comparisons**	**WT-HDM vs. WT-Sal**	**KO-Sal vs. WT-Sal**	**KO-HDM vs. WT-Sal**	**Overlap of Three Comparisons** **(Same Direction)**	**KO-HDM vs. WT-HDM**
**Bulk cell analysis**					
EpCAM+ cells	830	98	817	48 (36)	72
**Individual cell types**					
AT1	0	0	1	0	0
AT2	732	70	745	39 (30)	78
Alveolar progenitor (AP)	431	18	445	4 (2)	65
Broncho alveolar stem (BAS)	172	4	154	1(1)	51
Ciliated	210	41	200	5 (5)	0
Stromal	72	19	54	0	1
Club	55	5	88	0	0
Goblet	13	0	0	0	0
Neuroendocrine	88	3	16	0	0
**Total**	1773	160	1703	49 (45)	195

## Data Availability

The data have been deposited to the NCBI Sequence Read Archive (SRA), accession # PRJNA838071.

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
