# Peer review of "Single-Cell RNA-Seq Analysis Reveals Lung Epithelial Cell Type-Specific Responses to HDM and Regulation by Tet1"

_genes, 2022, doi:10.3390/genes13050880_

Round 1
Reviewer 1 Report
A brief summary
The authors performed single-cell RNA sequencing of lung tissue taken from a murine tet1 knock-out model (and WT) with and without house dust mite-induced allergic inflammation. The aim was to understand the role of tet1 in allergic inflammation in asthma on a cell-specific basis as this has not previously been described. The authors report cell-specific HDM-induced transcriptional changes of allergic inflammation as well as those induced commonly, and/or specifically by Tet1. Furthermore, the authors describe transcription factor activity and trajectory analysis.
General comments
The authors conduct a study that has appropriate controls and novel findings but overall, the presentation of the data can be improved. Figures were difficult to discern and figure 4 and figure 5 were completely unreadable. Furthermore, the presentation of key findings is somewhat obscured by the focus on genes that are commonly differentially expressed between cell types – why is this a major focus? Placing the HDM/Tet1-induced changes within their cell-specific and wider biological context should be of more prominent importance. Genes do not act in isolation so the primary focus on single gene-level changes here is not particularly insightful.
The best sections of the results are where genes are grouped by function and the function is provided in brackets. Please revise the presentation of the results to focus more explicitly on the wider function of the detected DEGs, as well as discussing results in the context of differences in cell states between conditions (the major strength of scRNAseq), this would improve the quality of the manuscript. The long sections detailing overlap in DEG between cell types and/or conditions provide little benefit and are provided concisely in Figure 3.
Although the authors find genuine differences in the Tet1 KO mice, these findings are obscured by the focus on single gene-level changes. To improve the quality of this publication the authors should explicitly propose, based on their findings, a model by which Tet1 may serve a protective role in HDM-induced lung inflammation. I.e. please expand on Figure 8 – how might Tet1 act to suppress these changes to protect against lung inflammation? In the conclusions section, the authors state that the AT2 cells are essential for Th2 inflammation – this pathway should also be added to figure 8. Furthermore, what about the role in ciliated cells? Is this achieved through activity on the TFs highlighted in the analysis? If so, what is the role of these TFs in these pathways? Little interpretation is given to the TF analysis results. For example, this is done for the AhR signaling pathway.
Specific comments:
“the mice were divided into 4 groups, with 2 mice in each group: WT-Saline, WT-HDM, KO-Saline, and KO-HDM” is in the supp but this should be brought into the main manuscript, it is an important detail.
It is impossible to read the gene names on the y-axis of S5 B,C,D. If it is intended that the reader know the gene names the authors will need to include fewer genes, alternatively if the cluster patterns between cell types is the only intention of the figure the authors should remove the gene names from the y-axis.
The reference numbers are not aligned to the correct references. For example, ref 17 is listed for scAlign but it is actually ref 14. Ref 14 is listed for Seurat which is not correct. There are many more cases of this.
URL for IPA also doesn't lead to a valid page.
Reviewer 2 Report
Authors should prepare a major revision in second review.
- Please resubmit the good quality figures!!!
- In result 3.1, authors showed that Tet1 mostly expressed in AT2 cell, alveolar progenitor cells, bronchial stem cells, ciliated cells and club cells. Please show the data.
- In result 3.2, they found 732 DEGs in AT2 cells, 476 were up and 256 were down between WT-HDM and WT-Sal. Is Tet1 DEGs?
- In result 3.3,there were 70 DEGs in AT2 cells, how many DEGs were they regulated by Tet1? And they also found 41 Tet1 loss-associated DEGs in ciliated cells. Were 41 DEGs in 70DEGs? If yes, how did authors identify 48 DEGs unique in AT2 cells?
- In result 3.4, please modify the tile 3.3-3.6 to 3.4-3.7!!! And authors found 39 DEGs shared in 3 different comparisons of AT2 cell. How many overlap DEGs did authors find between 39DEGs in 3.4 and 70 DEGs in 3.3? They also found 5 DEGs down regulated in ciliated cells. Are they belongs to 41 DEGs in result 3.3? If they have these overlap, how do authors describe the relationship between AT2 and ciliated cells?
- In result 3.7, authors showed their data supported that AT2 cells differentiated from AP and BAS cells. Did they found there were changes of some marker genes expression?
Round 2
Reviewer 2 Report
N/A